# Whole-exome sequencing reveals the origin and evolution of hepato-cholangiocarcinoma

Anqiang Wang[1,2], Liangcai Wu[1], Jianzhen Lin[1], Longzhe Han[3], Jin Bian[1], Yan Wu[4], Simon C. Robson[4], Lai Xue[5], Yunxia Ge[6], Xinting Sang[1], Wenze Wang[7] & Haitao Zhao[1]

Hepatocellular-cholangiocarcinoma (H-ChC) is a rare subtype of liver cancer with clinicopathological features of both hepatocellular carcinoma (HCC) and intrahepatic cholangiocarcinoma (iCCA). To date, molecular mechanisms underlying the co-existence of HCC and iCCA components in a single tumor remain elusive. Here, we show that H-ChC samples contain substantial private mutations from WES analyses, ranging from 33.1 to 86.4%, indicative of substantive intratumor heterogeneity (ITH). However, on the other hand, numerous ubiquitous mutations shared by HCC and iCCA suggest the monoclonal origin of H-ChC. Mutated genes identified herein, e.g., VCAN, ACVR2A, and FCGBP, are speculated to contribute to distinct differentiation of HCC and iCCA within H-ChC. Moreover, immunohistochemistry demonstrates that EpCAM is highly expressed in 80% of H-ChC, implying the stemness of such liver cancer. In summary, our data highlight the monoclonal origin and stemness of H-ChC, as well as substantial intratumoral heterogeneity.

[1] Department of Liver Surgery, Peking Union Medical College Hospital, Chinese Academy of Medical Sciences and Peking Union Medical College, Beijing 100730, China. [2] Department of Gastrointestinal Surgery, Key Laboratory of Carcinogenesis and Translational Research (Ministry of Education), Peking University Cancer Hospital & Institute, Beijing 100142, China. [3] Department of Pathology, Yanbian University Hospital, Yanji 133000, China. [4] Liver Center and The Transplant Institute, Department of Medicine, Beth Israel Deaconess Medical Center, Harvard Medical School, Boston, MA 02215, USA. [5] Department of Surgery, The University of Chicago Medicine, Chicago 60637, USA. [6] Novogene Bioinformatics Technology Co., Ltd, Beijing 100083, China. [7] Department of Pathology, Peking Union Medical College Hospital, Chinese Academy of Medical Sciences and Peking Union Medical College, Beijing 100730, China. Anqiang Wang, Liangcai Wu, Jianzhen Lin and Longzhe Han contributed equally to this work. Correspondence and requests for materials should be addressed to H.Z. (email: ZhaoHT@pumch.cn) or to W.W. (email: wwzvssxy@126.com) or to X.S. (email: sangxt@pumch.cn)

Primary liver cancer is one of the most common cancers worldwide[1] and the second leading cause of cancer-related mortality (745,000) every year[2]. Hepato-cholangiocarcinoma (H-ChC) is a special type of liver cancer with pathological features of both hepatocellular carcinoma (HCC) and intrahepatic cholangiocarcinoma (iCCA), accounting for 1–14.1% of primary liver cancers[3,4]. Risk factors for H-ChC include infection with the hepatitis B virus (HBV) or hepatitis C virus (HCV), alcohol consumption, and primary sclerosing cholangitis[3,5].

Coexistence of HCC and iCCA within H-ChC is an intriguing phenomenon. H-ChC was first reported by Allen and Lisa in 1949[6]. Accordingly, H-ChC was divided into three subtypes inclusive of separate masses composed of either HCC or iCCA/contiguous but independent masses of HCC and iCCA/an intermingling of hepatocellular and glandular elements. In 1994, WHO formally defined H-ChC as a tumor-containing intimate and unequivocal admixture of both HCC and iCCA[7]. HCC and iCCA are considered as two distinct types of liver cancer with different pathogenesis, pathological features, prognosis, as well as responses to adjuvant therapies. Up to date, the cellular origins of HCC and iCCA in H-ChC (viz. whether HCC and iCCA differentiate from the same cell origin or from distinct clones) and the underlying mechanisms remain largely unknown.

In this study, we performed whole-exome sequencing (WES) analysis using microdissected HCC and iCCA tissue specimens obtained from H-ChC patients to explore the relationships of these two H-ChC components. We find that H-ChC samples contain substantial private mutations as well as private somatic CNVs, indicative of intratumor heterogeneity (ITH) of H-ChC. However, large amount of ubiquitous nonsynonymous mutations and CNVs are overlapped in HCC and iCCA samples, suggesting the monoclonal origin of H-ChC. Mutated genes identified herein, e.g., *VCAN*, *ACVR2A*, and *FCGBP*, are speculated to contribute to distinct differentiation of HCC and iCCA within H-ChC. Moreover, EpCAM is highly expressed in H-ChC samples and associated with poor prognosis of patients with liver cancer. Hence, exome sequencing analyses are suggestive of monoclonal origin and stemness of H-ChC, as well as substantial intratumoral heterogeneity.

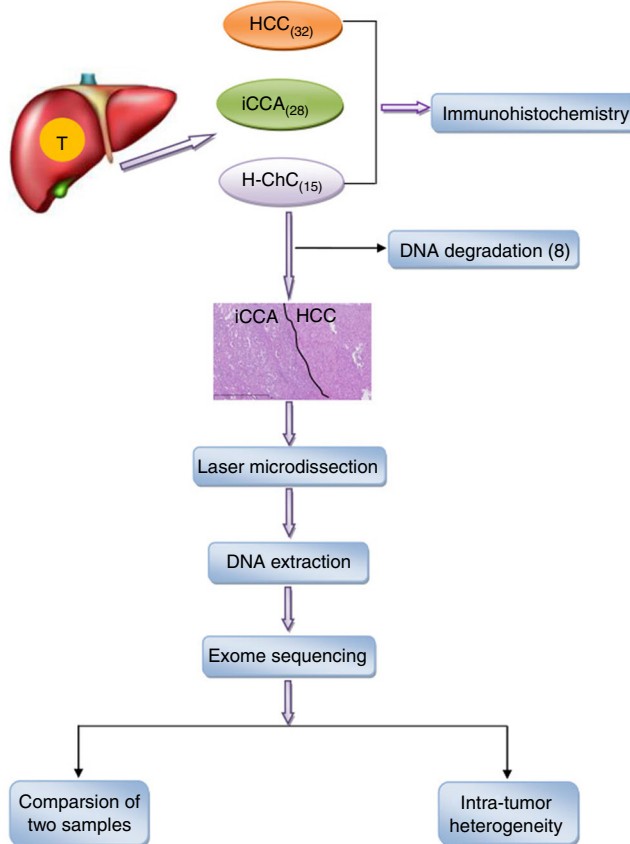

**Fig. 1** Flowchart of patient selection and experimental procedure. A total of 32 hepatocelluar carcinoma (H-ChC), 28 cholangiocarcinoma (iCCA), and 15 hepato-cholangiocarcinoma (H-ChC) were selected for immunohistochemistry. The blackline in the pathological picture of H-ChC separate HCC component from iCCA component in H-ChC. Finally, seven H-ChC patients were included for microdissection and DNA extraction. Then, we conducted whole-exome sequencing and exome data analysis

## Results

**Re-evaluation and pathological confirmation of H-ChC.** Figure 1 outlined the overall approaches that we used for this study. Tumor samples from 15 H-ChC patients were selected for immunohistochemical (IHC) staining. Tumor specimens from 32 HCC patients and 28 iCCA patients were used as controls. The clinical characteristics of all patients and the immunohistochemical profiles are summarized (Supplementary Data 1 and Supplementary Table 1), respectively.

As shown in Fig. 2a, b for expression of hepatocytic markers Hep and GPC3, vast majority of HCC and H-ChC samples showed positivity with Hep or GPC3 (93.8% for HCC and 80% for H-ChC contained in the HCC component). In contrast, only 14.3% of iCCA samples had hepatocytic expression.

In parallel, biliary differentiation was examined using the two standard markers CK7 and CK19. Hundred percent of iCCA samples exhibited biliary differentiation with positive staining for CK7 or CK19, and 93.3% of H-ChC samples showed such staining pattern only in the iCCA component (Fig. 2a, b). On the contrary, 25% of HCC samples showed biliary differentiation.

These pathological results are concordant with the clinical features of such different subtypes of liver cancer confirming the original diagnosis.

**Systematic mutation distribution in H-ChC patient samples.** Next, HCC, iCCA, and adjacent noncancerous tissues were

isolated from seven H-ChC patients using laser microdissection and subjected for WES. The average depth of WES ranged from 93.01× to 215.6× in all 21 samples (Supplementary Table 2). Somatic nucleotide variants (SNVs), indels, somatic copy number variants (CNVs), and HBV integrations were analyzed to first evaluate distribution of mutations in H-ChC samples.

On average, each H-ChC patient has 159 nonsynonymous SNVs, 6 frameshift indels, 136 CNVs, and 2.6 HBV integration sites (Supplementary Data 2 and 3 and Supplementary Table 3). Six mutational profiles were observed within tumor samples and the predominant mutational profile C:G>T:A is similar to that of liver cancer summarized by Alexandrov et al.[8]. Moreover, 70 known driver mutation genes inclusive of TP53, MTOR, and ARID2 were validated, all of which have been previously linked to liver cancer carcinogenesis (Supplementary Table 4). Among 359 significantly mutation genes (SMGs) identified in all samples, many genes, e.g., *FCGBP*, *ARID1B*, *PTPG13*, *GRM1*, and *MUC12*, have not been well studied in hepatocarcinogenesis (Supplementary Table 5). In addition, we also found 85 genes that may predispose people to liver cancer (Supplementary Table 6).

**Substantial intratumor heterogeneity in H-ChC.** To explore the relationship between HCC and iCCA components in the same H-ChC tumor, we performed whole-exome mutation and CNV analyses for all tumor samples. We classified somatic

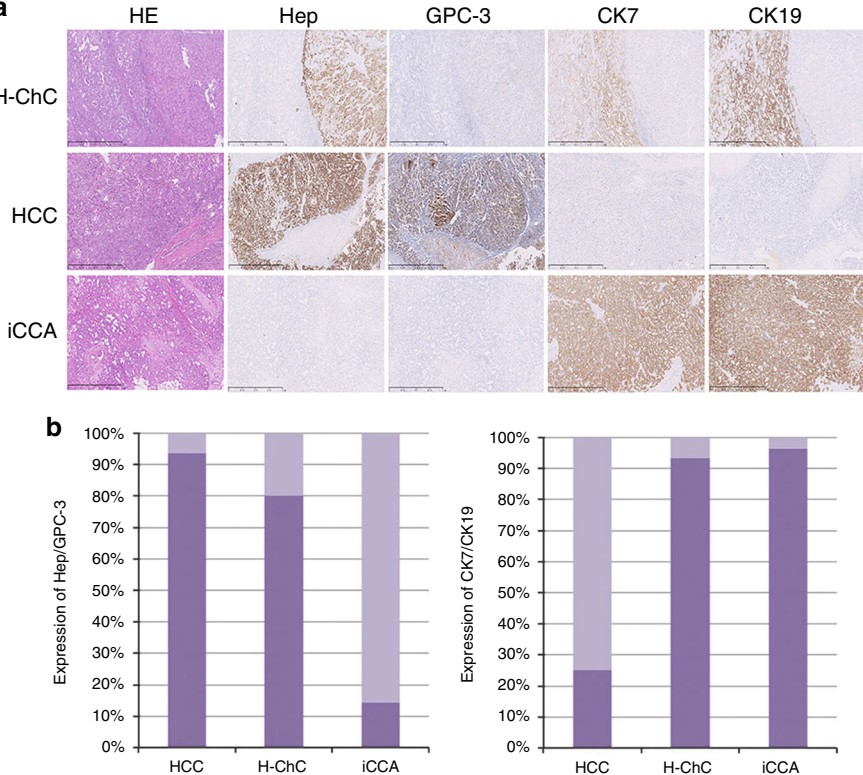

**Fig. 2** Immunohistochemical profiles of liver cancer. **a** The tumor cells show immunoreactivity for Heptocyte (Hep), GPC-3, CK7, and CK19 in H-ChC (P6). Immunocreativity for Hep and GPC-3 are observed in HCC. Tumor cells are positive for CK7, CK19 in iCCA. **b** The columns show the expression level of Hep/GPC3 and CK7/19 in H-ChC, HCC, and iCCA. The scale bar represents 1 mm

nonsynonymous mutations as ubiquitous (present in both tumor components) or private (exclusively present either in HCC or iCCA). As shown in Fig. 3, all H-ChC samples had substantial amount of private nonsynonymous SNVs, accounting for 86.4, 48.1, 77.3, 33.1, 56.6, 48.7, and 53.1% in patient 1 to patient 7 (P1–7), respectively.

Similarly, frequent private somatic CNVs were also noted in HCC or iCCA components ranging from 88.9 to 100% (Supplementary Table 7). These results suggest that substantive genetic heterogeneity exists in H-ChC, which is in keeping with the obvious morphologic differences between HCC and iCCA components seen in the clinic.

**HCC and iCCA components share common clonal origin in H-ChC.** On the other hand, ubiquitous nonsynonymous SNVs that are overlapped by HCC and iCCA samples range from 29 to 108 (Fig. 3). Moreover, mutation spectrum of iCCA is near identical to that of HCC within a given H-ChC (Fig. 4a). These data indicate that it is unlikely that HCC or iCCA component independently accumulated such substantial amount of ubiquitous nonsynonymous mutations.

Next, nonnegative matrix factorization approach was applied to identify mutational signatures. After cosine similarity analysis with 30 known signatures, three newly identified mutational signatures (A, B, and C) were found in H-ChC (Supplementary Fig. 1–3). For a given H-ChC sample, the percentage of mutations contained in each signature is consistent between HCC and iCCA (Fig. 4a). Additionally, HCC and iCCA share the same mutations on SMGs (Fig. 4d).

In parallel, ubiquitous somatic CNVs that are shared by HCC and iCCA were also identified. Number of ubiquitous somatic CNVs from P1 to P4 are 17, 2, 42, 36, and 5 for P7, respectively (Supplementary Table 9). However, no ubiquitous somatic CNVs

was found in P5 and P6. Within the same H-ChC sample, distribution of somatic CNVs in HCC is similar to that in iCCA (Fig. 4b, c and Supplementary Figures 4 and 5).

Furthermore, HBV integrations were also assessed. We observed that, in P5 samples, HCC and iCCA contain a common HBV integration site on chromosome 15 (Fig. 4c), suggesting that HBV integration is acquired before tumor differentiates into the two components. However, in P1, P4, and P6 samples, private HBV integrations were noted in either HCC or iCCA, indicating that HBV integrations could also occur after tumor differentiation into distinct phenotypes (Supplementary Figure 5). No HBV integration sites were identified in P2, P3, and P7 samples (Supplementary Figure 5).

To further explore the clonal architecture of H-ChC, we constructed phylogenetic trees (select driver genes were marked in the trunk and branch of evolutionary trees; Fig. 5a and Supplementary Table 8) using somatic SNVs and somatic mutations that take the CNVs in consideration. The results revealed different length of trunk in HCC and iCCA components of all H-ChC, which suggested the monoclonal origin of H-ChC (Fig. 5a, c). In addition, clonal analyses suggested common mutation clusters for all seven H-ChC patients after adjusting for cancer cell fraction (CCF) for each sample using Pyclone (Fig. 5b)[9–11].

The findings above obtained by analyzing systematic mutations, somatic CNVs, clonal analyses as well as HBV integrations all imply that H-ChC originates from a common progenitor.

**Genomic dynamics of two distinct differentiation phenotypes.** To further explore the genomic mechanisms of differentiation of monoclonal H-ChC into two distinct phenotypes, we screened all SMGs identified by WES. From 51 genes mutated in at least two H-ChC patients, we selected 17 genes associated with regulation

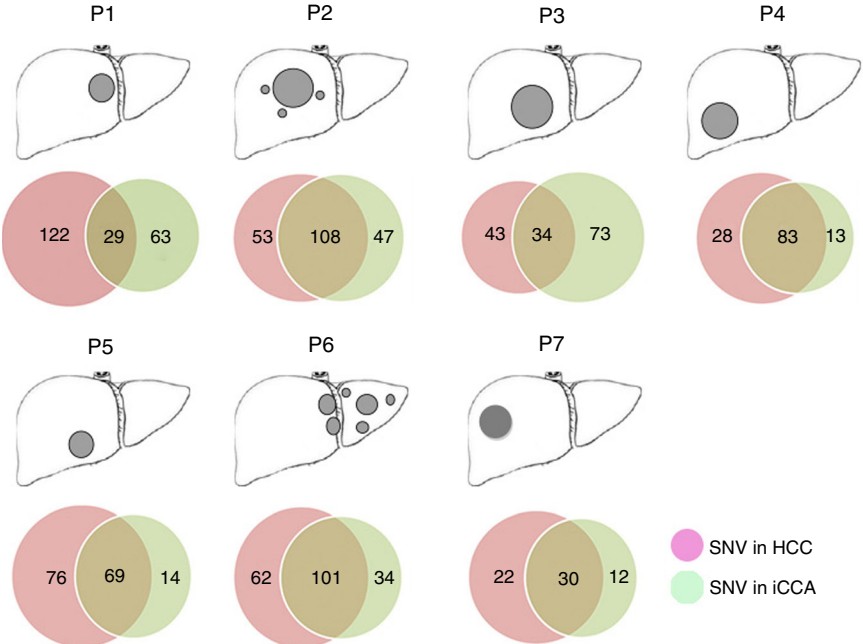

**Fig. 3** Distribution of nonsynonymous SNVs. **a** The cartoons of liver show the tumor sites in patients. **b** Venn diagrams show the relationship of nonsynonymous somatic mutations between H-ChC component (red circle) and iCCA component (green circle) in every H-ChC patient. The different number represents the mumber of nonsynonymous somatic mutations for corresponding samples and the overlapped regions are ubiquitous nonsynonymous somatic mutations between two samples of same H-ChC patients

of cell differentiation for subsequent pathway enrichment analysis (Supplementary Table 9). Mutations of nine genes, such as FCGBP and VCAN, that have been related to the maintenance of stem state were found in all seven H-ChC patients (Supplementary Table 9). Twelve genes, e.g., ACVR2A and APC, that constitute signaling pathways modulating pluripotency of stem cells were mutated in six H-ChC patients (P1–P6) (Supplementary Table 10). Moreover, 12 mutated genes in Wnt and Notch pathways including TP53 and NFATC2/3, which regulate the differentiation of hepatocyte and biliary epithelium, were noted in five H-ChC patients (P1–P4 and P6) (Supplementary Table 11). These data suggest the involvement of such key genes and pathway in the differentiation of HCC and iCCA in H-ChC.

**Expression of stem cell markers and survival analyses**. To further validate the stem features of H-ChC, we performed immunohistochemical staining with stem cell markers including c-kit and EpCAM in FFPE tissue specimens obtained from 75 liver cancer patients. Except for one HCC sample, all samples showed negative staining for c-kit (Fig. 6a). In contrast, vast majority of specimens exhibited EpCAM immunoreactivity (86.7% in H-ChC and 71.4% in iCCA) (Fig. 6b). Moreover, in HCC samples, EpCAM staining pattern appears to correlate with CK19 positivity (indicative of biliary differentiation), that is 66.7% in CK19 (+) samples vs. 17.2% in CK19 (−) samples (Fig. 6b).

We then conducted survival analysis by protein levels of EpCAM regardless of the cancer subtypes. Univariate analysis with log-rank test showed that EpCAM positivity was significantly associated with poor prognosis of liver cancer patients ($P = 0.028$; Table 1 and Fig. 6c). Yet, the prognostic value was not statistically significant as determined by multivariate Cox regression analysis ($P = 0.087$; Table 1). CA199 and margin status were found to be associated with prognosis of liver cancer by multivariate Cox regression analysis ($P = 0.001$; $P = 0.015$; Table 2).

## Discussion

H-ChC is a special type of liver cancer that contains both HCC and iCCA components. Herein, we showed that hepatocytic markers (Hep and GPC3) were strongly expressed in the HCC component of H-ChC, whereas biliary epithelial markers (CK7 and CK19) were strongly expressed in the iCCA component of H-ChC. We then conducted WES to explore the relationship between the two distinct components of H-ChC. We found that HCC and iCCA components possessed many private nonsynonymous mutations and somatic CNVs. Although we conducted CNV analysis using exome data, this method proved to be highly accurate compared to the results of CNV analysis using low-depth whole genome data[12]. Private somatic CNVs occurred more frequently than private somatic mutations. These findings support the notion that substantial intratumor heterogeneity exists within H-ChC. Intratumor heterogeneity in H-ChC poses a great challenge for liver cancer-targeted therapy[12–14]. Currently, it is a big challenge to choose the target drugs from multiple mutated driver genes. However, the common mutations within H-ChC may be a better choice because of their more important role in carcinogenesis. Therefore, it is necessary to conduct microscopic multiregional tissue selection in order to understand the complete genetic landscape of H-ChC, which can aid in the development of targeted molecular therapies.

On the other hand, the prevalence of ubiquitous somatic SNVs (29–108) and CNVs (2–42) noted in H-ChC suggests that HCC and iCCA components of H-ChC share a common progenitor. The relatively narrow coverage for genomic of WES makes it difficult to acquire all of the CNVs. However, the identification of ubiquitous somatic CNVs could also support the monoclonal origin of H-ChC to some extent. And the lack of ubiquitous somatic CNVs could not rule out the possibility of H-ChC monoclonal origin. Because the likelihood of HCC and iCCA component independently accumulating such substantial amount of ubiquitous nonsynonymous mutation is exceedingly low. Clonal analyses using Pyclone identified common mutation

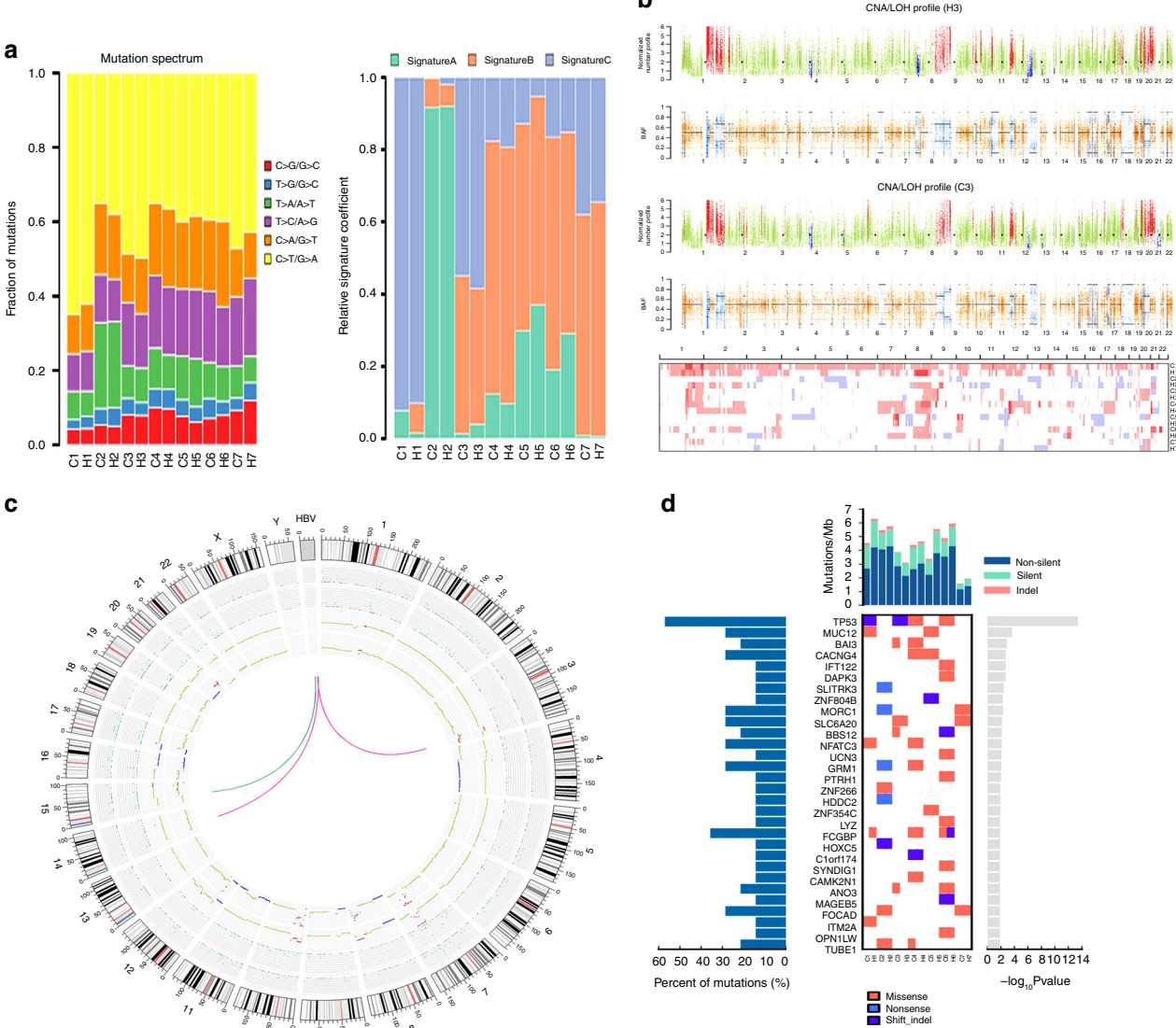

**Fig. 4** Mutation spectra and mutation signatures among H-ChC samples. **a** The upper column depicts mutation spectrum within all tumor samples. Red represents mutation type C>G/G>C, blue T>G/A>C, green T>A/A>T, purple T>C/A>G, orange C>A/G>T, and yellow C>T/G>A. Y axis indicates fraction of matations. The below column shows mutation signatures of all tumor samples. **b** The upper chart depicts the distribution of somatic copy number variation (CNV) of patient 3 (P3). Red is for CNV gain, green for normal CNV, and blue for CNV loss. The below chart depicts the distribution of B allele frequency (BAF). Orange represents consistent distribution of allele and blue indicates loss of heterozygosity (LOH). Heat maps show the distribution of CNV for all included H-ChC patients. Red is for CNV gain and blue for CNV loss. **c** Circos plot depicts the relationship between HCC and iCCA components of P5 on the terms of somatic nucleotide variant (SNV), somatic insertions and deletions, somatic copy number variation (CNV), and HBV integration. The first and second circles represent CNV for iCCA and HCC. Red indicates CNV gain, green normal CNV, and blue CNV loss. The third and fourth circles represent SNV and indel for iCCA and HCC. Green dot indicates SNV and indel. Green curve indicates HBV integration sites for iCCA and red curve for HCC. **d** Significantly mutated genes (SMG) landscape shows the distribution of some SMGs between samples in H-ChC. The column on top shows the mutational rate of every sample. Heat map shows the SMGs and mutation type including missense mutation (red block), nonsense mutation (light blue block), and shift indel (dark blue block). H1 and C1 represent the HCC and iCCA components of patient 1 (P1). Similar label was used for other patients. The column on the left stands for the percent of mutations for SMGs. The picture on the right shows the P value for SMGs

lusters in both of tumor components, which suggested monoclone of H-ChC. For each H-ChC sample, the probability of monoclonal and bicolonal is equal, just as the two sides of coin. The probability of same clonal features for different samples decreases 50% with one sample increase, especially all samples were chosen randomly. Therefore, although seven H-ChC samples are small, the conclusion is also reliable. HBV integrations were found in four out of seven patients. A unique HBV integration site common to both HCC and iCCA component was found in P5. Considering the random features of HBV integration

and tens of millions of probable integration sites, the probability of common HBV integration sites in two different origin liver cancer samples could hardly be ignored. Therefore, the common integration supports the monoclonal origin of HCC and iCCA components. However, a lack of common HBV integration site in other samples cannot rule out the possibility of common origin of these two tumor components, especially in light of the data from WES analyses, because HBV could integrate in any sites of tumor genome. Furthermore, the HBV integration randomly occurs in any stage of carcinogenesis, either before or after differentiating

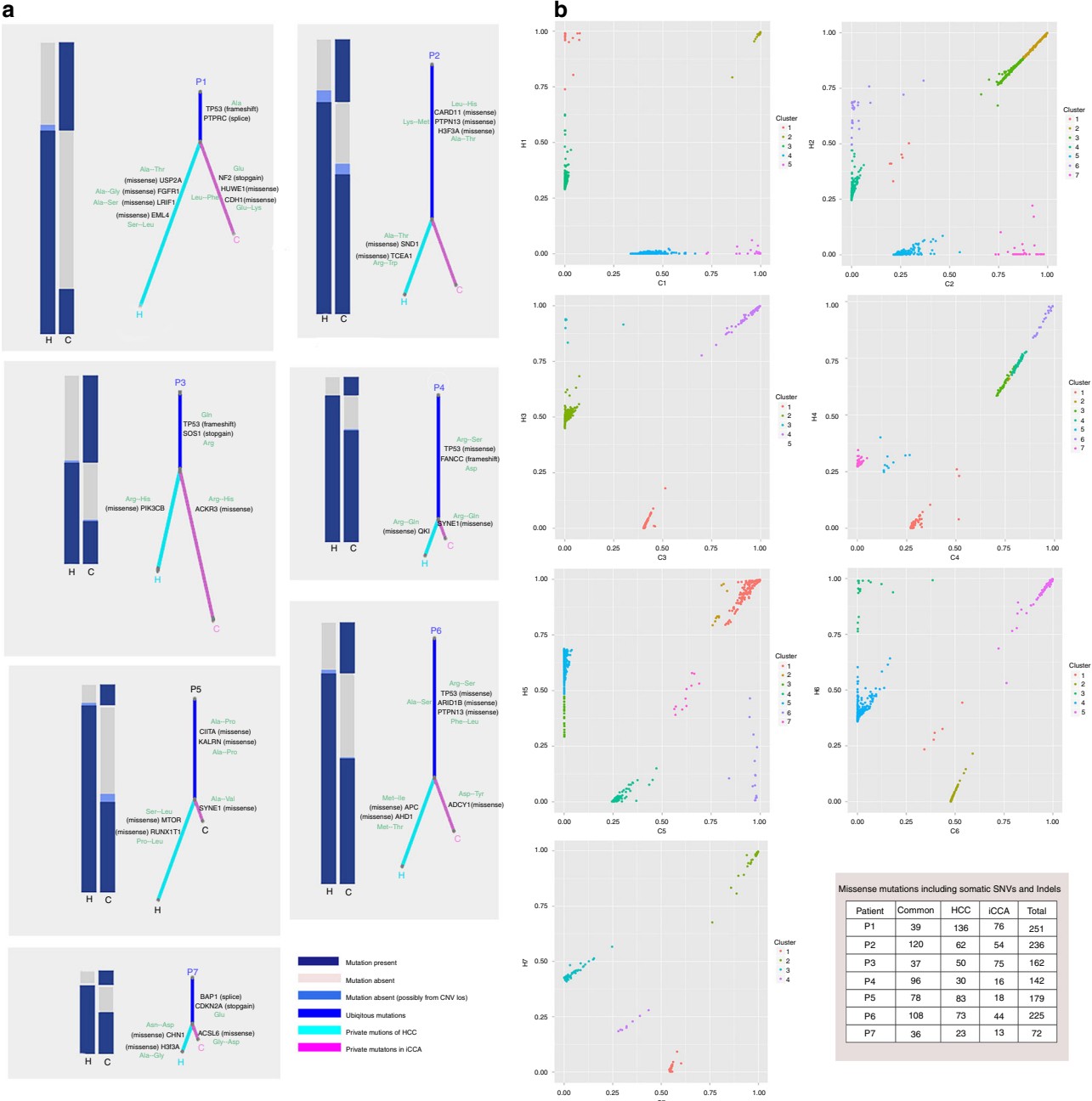

**Fig. 5** Missense mutations and cancer cell fraction comparisons within each H-ChC. **a** Heatplot with different bars represents various distributions of missense mutations including somatic SNVs and Indels within H-ChC. Fraction of ubiquitous nonsynonymous somatic mutations (trunk) (blue bar) and unique nonsynonymous somatic mutations (branch) (green bar for HCC and pink bar for iCCA) reveal the relationship of two tumor samples within a single H-ChC. Part of driver mutations was marked on trunk and branch of evolutionary trees. **b** Two-dimensional scatter plots show the cancer cell fraction (CCF) of the mutations in HCC and iCCA components of tumors. Different clusters were calculated from each H-ChC sample. Clusters off the axes indicate mutations in both of tumor components. Clusters on the axes reveal mutations in either HCC or iCCA components. **c** The table shows missense mutations in different tumor components

into distinct phenotypes. Indeed, it is not rigorous to identify HBV integration using WES on FFPE samples. However, we just regard the HBV integration as a supplementary evidence to explore the clonality of H-ChC.

Another puzzle is that how monoclonal H-ChC could become a tumor with unambiguous HCC and iCCA components. We hypothesize that some key genes or pathways may contribute to the loss of the undifferentiated state of stem cells and/or hepatocyte or biliary epithelial differentiation. WES analyses showed that many genes participated in cell stemness and differentiation

are mutated. For example, VCAN is known to favor homeostasis of extracellular matrix, thereby modulating cell proliferation and differentiation[15]. It is also essential for the maintenance of stem state of cancer stem cells in many types of cancer[16]. ACVR2A is overexpressed in undifferentiated states of mouse and human embryonic stem cell lines and decreased upon differentiation. All seven H-ChC patients possess mutated genes involved in cell stemness and differentiation. Mutated genes identified in six patients are the key components of signaling pathways regulating pluripotency of stem cells. In addition, Wnt and Notch pathways

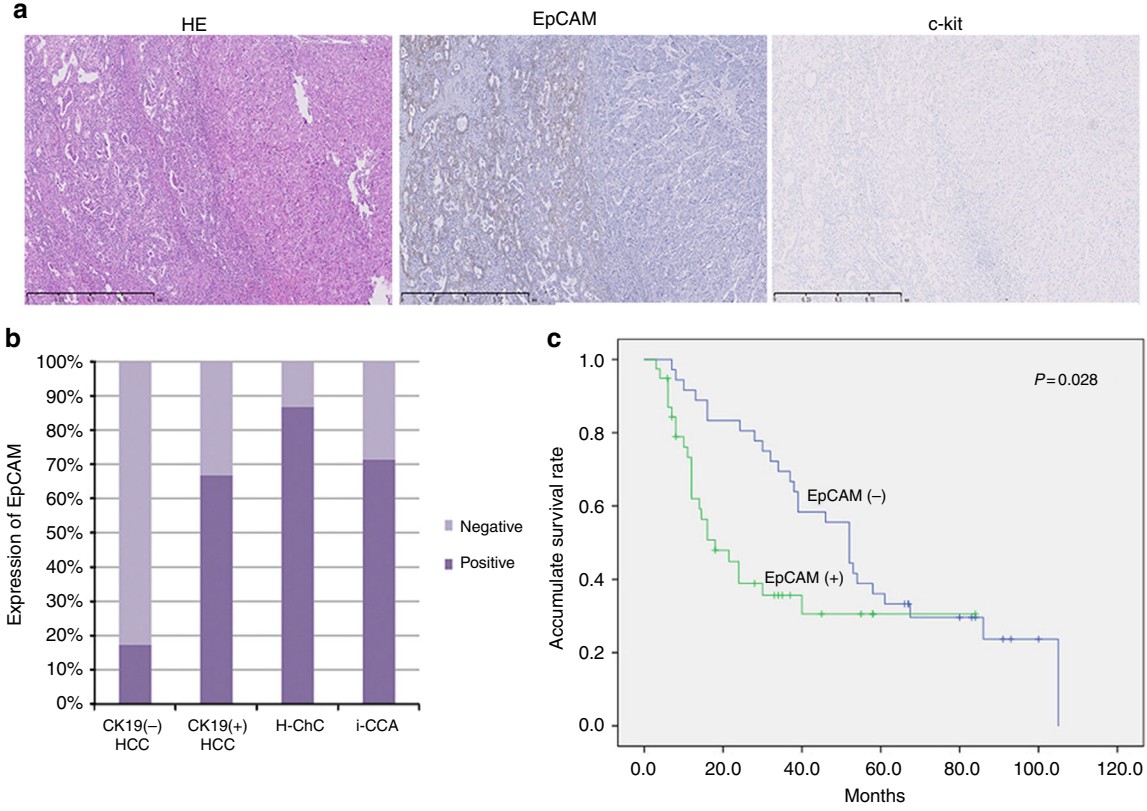

**Fig. 6** Expression of stem cell markers and survival analysis. **a** The tumor cells show immunoreactivity for EpCAM in H-ChC. **b** The columns show the expression level of EpCAM in H-ChC, HCC, and iCCA. **c** The survival curve shows the relationship between EpCAM expression and suivival. The scale bar represents 1 mm

have been known to modulate the differentiation of hepatocytes or billilary epithelial cells[17]. In this study, we also found mutated genes in these two pathways in five out of seven H-ChC patient samples. Overall, we demonstrated that some gene mutations might lead to the instability of stem state, whereas others contributed to distinct differentiation of H-ChC. However, whether there are any corresponding changes in the protein levels of such mutation genes remain unknown, which could be exploited using next-generation RNA sequencing.

Furthermore, we searched NCBI database to examine the relationships between mutation genes and well-known stem cell markers. However, no gene mutations in stem cell markers, e.g., EpCAM, NCAM, c-kit, CD133, and CD90, were found in any samples used for this study. We then analyzed protein levels of select stem cell markers inclusive of EpCAM and C-kit by immunohistochemistry using samples from 75 liver cancer patients. C-kit, considered as a marker for pluripotent stem cell in the liver, has previously been shown to characterize H-ChC, denoting the stem cell origin of H-ChC[18–20]. C-kit protein expression, however, was absent in nearly all liver cancer samples in our study. The other stem cell marker, EpCAM, is originally identified on the surface of embryonic stem cells[21] and has been shown to play a role in the maintenance of the undifferentiated state of stem cells[22,23]. Overexpression of EpCAM has been previously noted in H-ChC patients[18], which is in accordance with our results herein. Nevertheless, we are still not clear with regard to the origin of H-ChC, as stem cell-like features could derive either directly from stem cells or through reprogramming from mature differentiated liver cancer cells[24–26].

It is widely accepted that HCC can be classified into two subtypes according to the expression of biliary differentiation marker CK19 and CK19+ HCC has a worse prognosis than CK19−

HCC[27]. In this study, we observed that expression of EpCAM is markedly increased in CK19+ HCC samples when compared to CK19− samples, although CK19+ subtype only accounts for 25% of HCC. In parallel, majority of iCCA and H-ChC samples, those are chiefly CK19+, exhibited positivity for EpCAM. These results are in keeping with the previous studies, suggesting that CK19+ HCC and iCCA could originate from progenitor cells or biliary tree stem cells[26–29]. Moreover, by performing survival analyses, we demonstrated that liver cancer patients with high levels of EpCAM expression have poor prognosis using the univariate Cox analysis model. However, when using multivariate Cox analysis, the prognostic value of EpCAM expression was not statistically significant, possibly due to the limited sample size in our study.

Collectively, we found that the HCC and iCCA components of H-ChC originate from a common cell with stem cell-like features. This notion is supported by many other studies showing the stem cell origin of H-ChC[19,20,30]. Indeed in 2010, WHO updated the classification criteria for H-ChC: H-ChC is divided into the classical type and three subtypes with stem cell features (typical subtype, intermediate cell subtype, and cholangiolocellular subtype)[31]. Accordingly, all seven H-ChC patients subjected for exome analyses herein can be classified as the classical type. However, such WHO classification method is suboptimal having many issues that need to be urgently addressed. Many newly recognized different types of liver cancer have been suggested to be classified as a new subtype of H-ChC. For example, in 2013, Terada et al. reported an H-ChC case with ductal plate malformation features that was characterized by formation of markedly irregular tubules of iCCA cells with intraluminal cell projections, bridge formations, and intraluminal tumor biliary cells[32]. Jung et al. reported cholangiocellular carcinoma with satellite nodules showing intermediate differentiation[17]. In

## Table 1 Results of univariate analyses in liver cancer

| Variables | No. of pat. | OS | | | |
|---|---|---|---|---|---|
| | | No. of event | MST | $X^2$ | P |
| Gender | | | | | |
| M | 57 | 38 | 34.379 | 1.362 | 0.243 |
| F | 18 | 13 | 49.279 | | |
| HBV | | | | | |
| Yes | 47 | 31 | 50.225 | 0.955 | 0.328 |
| No | 28 | 20 | 37.103 | | |
| AFP | | | | | |
| Normal | 41 | 27 | 46.332 | 0.167 | 0.683 |
| Elevated | 28 | 21 | 43.324 | | |
| NA | | | | | |
| CA199 | | | | | |
| Normal | 38 | 24 | 49.987 | 8.404 | 0.004 |
| Elevated | 23 | 20 | 25.283 | | |
| NA | | | | | |
| Histology | | | | | |
| HCC | 32 | 20 | 61.16 | 39.516 | 0.009 |
| ICC | 28 | 21 | 32.858 | | |
| CHC | 15 | 10 | 29.038 | | |
| Size | | | | | |
| <5 cm | 43 | 29 | 51.091 | 1.095 | 0.295 |
| >5 cm | 32 | 22 | 40.582 | | |
| Number | | | | | |
| 1 | 52 | 32 | 51.091 | 1.095 | 0.295 |
| >2 | 23 | 19 | 37.885 | | |
| Differentiation | | | | | |
| High | 11 | 5 | 65.519 | 2.642 | 0.104 |
| Low/ modreate | 53 | 38 | 44.269 | | |
| Vascular invasion | | | | | |
| Yes | 9 | 7 | 36.820 | 0.778 | 0.378 |
| No | 66 | 44 | 46.773 | | |
| Margin | | | | | |
| Positive | 15 | 13 | 51.995 | 7.732 | 0.005 |
| Negative | 60 | 38 | 23.267 | | |
| TACE | | | | | |
| Yes | 44 | 31 | 45.507 | 0.073 | 0.788 |
| No | 31 | 20 | 47.191 | | |
| Hepatocyte | | | | | |
| Positive | 39 | 25 | 54.548 | 3.711 | 0.054 |
| Negative | 36 | 26 | 35.337 | | |
| GPC3 | | | | | |
| Positive | 32 | 23 | 45.197 | 0.104 | 0.747 |
| Negative | 43 | 28 | 47.373 | | |
| Hepatocyte/GPC3 | | | | | |
| Positive | 47 | 30 | 52.426 | 2.918 | 0.088 |
| Negative | 28 | 21 | 34.572 | | |
| CK7 | | | | | |
| Positive | 43 | 29 | 39.425 | 2.3 | 0.129 |
| Negative | 22 | 22 | 54.523 | | |
| CK19 | | | | | |
| Positive | 42 | 28 | 35.361 | 4.698 | 0.030 |
| Negative | 33 | 23 | 56.665 | | |
| CK7/CK19 | | | | | |
| Positive | 49 | 33 | 39.721 | 2.670 | 0.102 |
| Negative | 26 | 18 | 56.772 | | |
| EpCAM | | | | | |
| Positive | 39 | 24 | 36.206 | 4.841 | 0.028 |
| Negative | 36 | 27 | 56.132 | | |

No. of pat. number of patient; OS, overall survival; MST, mean survival time; TACE, transcatheter arterial chemoembolization

## Table 2 Results of multivariate analyses for liver cancer

| Variables | OR | 95% CI | P |
|---|---|---|---|
| CA199 | 4.156 | 1.769–9.766 | 0.001 |
| Histology | 1.430 | 0.816–2.503 | 0.211 |
| Size | 1.291 | 0.602–2.767 | 0.512 |
| Number | 1.214 | 0.585–2.518 | 0.603 |
| Vascular invasion | 1.100 | 0.382–3.169 | 0.860 |
| Margin | 2.874 | 1.227–6.773 | 0.015 |
| TACE | 2.221 | 1.029–4.796 | 0.042 |
| CK19 | 2.212 | 0.717–6.826 | 0.618 |
| EpCAM | 2.151 | 0.896–5.165 | 0.087 |

OR, odds ratio; 95% CI, 95% confidence interval

tumor consists of plural components, which we also adopted. Up to now, however, no widely accepted standard is adopted. Furthermore, the classical type of H-ChC is proved to be monoclonal. Therefore, even if they are not positive for widely used stem markers, they may also originate from cells with stem features. In this case, we assumed that classical type of H-ChC also belong to H-ChC with stem cell features. Herein, we suggest a subgroup classification strategy based on the cell origin rather than morphology features, e.g., EpCAM+ H-ChC. We speculate that other types of H-ChC, CK19 (+) HCC, and part of iCCA all belong to this type of cancer. Of course, further studies are needed (e.g., by using next-generation sequencing analyses) to validate the clinical utility of such novel classification recommendation.

In conclusion, exome sequencing analyses are suggestive of the monoclonal origin of H-ChC, which may promote the molecule classification of primary liver cancer on the basis of cell origin. In addition, the substantial intratumor heterogeneity noted in H-ChC urges multiregional sequencing analysis to find the common driver mutations that playing more important role in carcinogenesis, thus make target drugs selection more precision and effective.

## Methods

**Patients and tissue samples.** We enrolled 75 patients who underwent surgery at Peking Union Medical College Hospital from April 2008 to December 2015. Fifteen patients were pathologically diagnosed with combined hepatocellular cholangiocarcinoma (H-ChC), 32 were diagnosed with hepatocellular carcinoma (HCC), and 28 with intrahepatic cholangiocarcinoma (iCCA). All histological specimens were reevaluated by two experienced pathologists. HCC, iCCA components of H-ChC samples, and adjacent non-cancerous liver tissue were collected from seven eligible H-ChC patients. The flowchart of patients selected for DNA extraction and whole-exome sequencing was presented (Fig. 1). Fifteen H-ChC were performed immunohistochemistry. Thirty-two HCC and 28 iCCA were also performed immunohistochemistry as control group. Informed consent was obtained from all human participants. This research was also approved by the Ethical Committee of Peking Union Medical University Hospital. All clinical data were obtained from hospital record for each case (Supplementary Data 1).

**Immunohistochemistry.** We cut 5-μm-thick sections from representative paraffin blocks of liver cancer for immunohistochemistry (Supplementary Figure 6). The primary antibodies were hepatocyte (IR624, 1:50, Dako), GPC-3 (ZM-0146, 1:1, ZSGB-Bio), CK7 (1:1, Dako), CK19 (1:50, Dako), EpCAM (HEA125, 1:50, abcam, Cambridge, UK), and c-kit (A4502, 1:1, Dako). Immunoreactivity was evaluated according to the percentage of positive cells on sections regardless of intensity of staining, with grading form 0 to 4+ as follows: 0, no staining or equivocal reaction; 1+, 1–5% positive cells; 2+, 6–25% positive cells; 3+, 26–50% positive cells; and 4+, >50% positive cells. Tumors with a grade of 2+ were regarded as positive for antigen expression.

**Microdissection and DNA extraction.** Deparaffinized tumor specimens were cut into serial consecutive 10 μm slides, stained with hematoxylin. Two pathologists confirmed the different cancer components independently. Then, HCC and iCCA components were separately microdissected from H-ChC slides using Leica CTR 6000 Microsystem (Wetzlar, Germany). We also dissected adjacent nonmalignant

addition, a single H-ChC tumor often contains more than three morphologically distinct tumor components and classification of these atypical tumors remains problematic. Some research[18] made diagnosis by the predominant (50%) components when

tissue as control tissue. Then, DNA was extracted using the QIAamp DNA FFPE Tissue Kit Print (Qiagen, Germany) according to the manufacturer's protocol.

**Library preparation and sequencing**. Paired-end DNA library was prepared according to the manufacturer's instructions (Agilent). The adapter-modified gDNA fragments were enriched by six cycles of PCR. Whole exome capture was carried out using Agilent's SureSelect Human All Exon V5 Kit. Finally, 50 Mb of DNA sequences of 33,4378 exons from 20,965 genes were captured. After DNA quality evaluation, pooled samples were sequenced on Illunima Hiseq 4000 according to the manufacture's instructions for paired-end 150 bp reads. The average sequencing depth of target region and coverage of target region were summarized in Supplementary Table 2.

**Exome sequencing data analysis for SNVs and INDELs calling**. Raw data (stored as FastQ format) obtained from Hiseq4000 contains adapter contamination, low-quality nucleotide, and undetected nucleotide (N), which can pose significant influence on downstream processing analysis. Hence, reads with adapter contamination, reads containing uncertain nucleotides more than 10 percentage, and paired reads when single reads have more than 50 percentage low-quality (<5) nucleotides are discarded. After these steps, high-quality clean data are obtained. Finally, QC statistics including total reads number, sequencing error rate, percentage of reads with average quality >Q20, percentage of reads with average quality >Q30, and GC content distribution can be calculated. Paired-end clean reads are aligned to the reference genome (UCSC hg19) using Burrows–Wheeler Aligner (BWA) software[33]. If a read or reads pair is mapped to multiple positions, BWA will choose the most likely placement. While if two or more most-likely placements are present, BWA will choose any one randomly. Aligned reads were realigned to the genome. Genome Analysis Toolkit (GATK)[34] was used to ignore those duplicates resulted from PCR amplification with Picard-tool. We utilized the Indelrealigner and RealignerTargetCreator in GATK do realignment around the indels according to GATK best practice. Furthermore, we performed base quality score recalibration with GATK to avoid system bias. After realignment to genome, we identified and filtered variants (SNP, INDELs) using GATK HaplotypeCaller and variantFiltration to guarantee meaningful analysis. Variants obtained from previous steps were compared based on the dbSNP[35] and 1000 Genomes database[36] and annotates with ANNOVAR[37]. SNVs and somatic INDELs were identified using MuTech[38] and Strelka[39] with matched adjacent non-cancerous samples, respectively (Supplementary Data 2). The comparison of somatic mutations was presented in circus plot within two tumor samples in same patient[40].

**Mutation spectrum and mutation signature analysis**. We performed mutation spectrum and signature analysis to explore the relationship within tumor samples in each same patient. We drew mutation spectrum barplot to present mutation spectrum of each sample. We also conducted clustering analysis on mutation spectrum to draw mutation spectrum heatplot to observe the similarity and difference within tumor samples. We conducted cluster analyses on 96 somatic mutation types using nonnegative matrix factorization[41,42] and acquired three different mutation signatures. Then, mutation spectra were clustered with 30 known signatures on COSMIC[43] to explain mutation process of samples[44]. The similarity of mutation signatures was evaluated with cosine similarity >0.9, which suggested common signatures. Signatures A, B, and C were identified in tumor samples and the distribution of them was presented.

**Estimation of tumor purity and cellularity**. We used ABSOLUTE to estimate the tumor purity and cellularity from analysis of somatic DNA alteration[45] (Supplementary Table 12).

**Phylogenetic and clonal analyses**. We conducted phylogenetic analysis based on the exome sequencing data. Branch and trunk lengths are proportional to the number of nonsynonymous mutations acquired on the corresponding branch and trunk[46]. Driver mutational genes were marked on trunk and branch. In order to evaluate the clonality of tumor more precisely, we conducted clonal analysis using Pyclone[47].

**Identification of potential driver mutations**. In order to identify potential driver mutations playing significant role in carcinogenesis, we compared sample mutations with known driver mutations with in-house software. Four driver mutation databases were utilized for the comparison, including Cancer Gene Census (CGC513)[48], Bert Vogelstein125[49], SMG127[50], and Comprehensive 435[51] database. Potential driver mutations were analyzed to determine common driver mutations for each patient (Supplementary Table 4).

**Determination of significantly mutated genes**. Significantly mutated genes (SMGs) were defined as the somatic mutations with higher mutated frequency than background mutation rate. We analyzed mutations with SMG test[52] based on tumor samples. Significantly mutated genes landscape heatmap was presented. For SMGs, we conducted pathway enrichment analyses with PathScan software[53].

Databases such as KEGG[54], Biocarta, PID, and Teactome were utilized to perform this analysis (Supplementary Table 5).

**Screening of predisposing genes**. Potential predisposing genes could be identified by detecting germline mutations and comparing with CGC[46] database with in-house software, which were presented in Supplementary Table 6.

**Copy number analysis using exome-sequencing data**. We identified copy number variants (CNVs) with control-FREEC[55] based on exome-sequencing data to analyze the copy number state of each tumor. Control-FREEC could construct copy number profiles with aligned BAM data. Then, the profiles were normalized, segmented, and analyzed to obtain the copy number state of each genomic region. We compared cancer tissues with matched non-cancerous tissues with control-FREEC. Tumor samples with overdiploidy could also be analyzed. We also performed comparison of CNVs between two tumor samples in same patient. We defined a common CNV if 100% of CNV regions of two samples overlap[56] (Supplementary Data 3 and Supplementary Table 7). The comparison of CNVs was presented with circular visualization[38] within two tumor samples in same patient.

**HBV integration analyses based on exome-sequencing data**. To identify the HBV integration sites based on exome-sequencing data, we constructed an HBV virus genome library. First, we conducted reads alignment to human reference genome and some soft-clipped reads were extracted. Then, mapping to human and virus reference genome were conducted. The HBV integration sites would be detected by clipping reveals structure (CTEST)[57] using an assembly-mapping-searching-assembly-alignment procedure. We explored the common HBV integration sites between HCC and iCCA components within same H-ChC samples.

**Data availability**. Sequence data have been deposited at the European Genome-phenome Archive (EGA), which is hosted by the EBI and the CRG, under accession number EGAS00001002783. All the data are available within the article, supplementary information, and supplementary data file, or available from the authors on request.

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

## Acknowledgements

We thank Elsevier Language Editing Services for their help in English language revision of this manuscript. We also thank for the support of CAMS Innovation Fund for Medical Science (CIFMS) (2017-12M-4-003), International Science and Technology Cooperation Projects (2015DFA30650 and 2016YFE0107100), Capital Special Research Project for Health Development (2014-2-4012) and Beijing Nature Science Foundation for Young Scholars Project (7164293).

## Author contributions

A.W. designed the study and wrote the manuscript. L.W. and J.L. analyzed exome-sequencing data. J.B. collected all samples. L.X. conducted the mamuscript revision. L.H. and W.W. conducted pathological diagnosis and immunohistochemistry. Y.W. and S.C.R. reviewed and edited this manuscript. Y.G. helped further analyze the sequencing data. X.S. and H.Z. co-ordinated and provided financial support for this work.

## Additional information

**Competing interests:** The authors declare no competing interests.

