## [Peer Review File · Nature Communications]

Reviewers' comments:

Reviewer #1 (Remarks to the Author):

The manuscript (NCOMMS-16-30595-T) is a well-executed manuscript examining mixed hepatocellular-cholangiocarcinoma (H-ChC) genetics. The authors appropriately employ microdissection followed by whole-exome sequencing of the samples. The authors report on differences and similarities on nonsynonymous mutations, somatic copy number variations, and HBV integrations between iCCA, HCC, and H-ChC cancers. The data are novel, contribute to our understanding of these cancers, and the data largely support the conclusions. I have several comments to strengthen the manuscript.

MAJOR COMMENTS:

1. The immunohistochemistry phenotyping of the cancers is incomplete. For HCC, in addition to GPC3, HSP70 and glutamine synthesis should be employed (EASL guidelines Journal of Hepatology 2012). For intrahepatic cholangiocarcinoma, SOX9 should be employed as an additional marker.
2. The current term for mixed cancers is now Hepato-Cholangiocarcinoma (H-ChC) as defined by the current mixed tumor working group.
3. The current term for intrahepatic cholangiocarcinoma is iCCA (see review by Blechacz et al. Nature Reviews in Gastroenterology and Hepatology 2011).
4. The Kaplan-Meier curve should be deleted from the manuscript. The numbers are too small, and the clinical information to assure equal stratification across the groups is not provided. Hence, all data on patient survival need to be removed from the manuscript.
5. Telomerase promoter mutations are common in early HCC (Nault JC et al. Nature Genetics 2015). What percent of the mixed tumors had these mutations?
6. By an Ingenuity-like Pathway Analysis, were there common pathways driving the differing phenotypes?

Reviewer #2 (Remarks to the Author):

The authors sequenced 15 combined hepatocellular cholangiocarcinoma, 32 HCC and 28 ICC. WES was performed with DNA extracted from FFPE samples with or without microdissection. Following these analyses, the authors concluded that combined tumors were of clonal origin with substantial intra-tumor heterogeneity.

This is an interesting project, however the manuscript and the experimental design showed major weaknesses and the conclusions are not innovative enough. Here are some selected examples:

1. Pathological features of these series of samples are not precisely described and this is a major weakness
2. Several clinical data are missing
3. The identified mutations are not sufficiently described in term of putative protein and functional consequences. Allele frequency and coverage should be provided. Consequently the results are very difficult to evaluate with only a list in pdf. Several genes classically mutated in HCC and ICC were not identified in the present work suggesting technical problems in the workflow
4. Nucleotide mutational signatures should be assessed according to COSMIC nomenclature
5. The text is very difficult to read and to understand.

Reviewer #3 (Remarks to the Author):

The authors perform a comparative analysis of laser micro-dissected hepatocellular carcinoma (HCC) and cholangiocarcinoma (ICC) components from seven patients diagnosed with combined

hepatocellular cholangiocarcinoma, a rare subtype of liver cancer that displays both pathological features. They perform whole exome sequencing to characterize the subclonal structure and determine the relationship between the two components. Although the study contains only seven patients and is underpowered, this is an interesting sample set. They identify common as well as sample-specific somatic alterations. They conclude that the two tumor compartments are of monoclonal origin with the accumulation of substantial sample-specific mutations in the form of branching evolution. Although the inference of the monoclonality is probably true, there are major flaws in experimental setup as well as the analysis that prevent a true assessment of the level of heterogeneity.

1. No information about the physical proximity of the HCC and ICC components is given. Were the samples taken from a single neoplastic mass that contiguously presents HCC and ICC or from physically different foci separated by normal-looking liver tissue?
2. Is there a pathologist's estimate of tumor cellularity? Was cellularity estimated from NGS data? Low tumor purity might affect the identification of mutations in the two histological compartments especially if the coverage is not comparable. Mutation calling algorithms might not be able to identify mutations if the variant allele fraction is low. Have the authors performed a pileup of sample-specific mutations in the other sample to see if there are any aberrant reads supporting the same variant?
3. Study of clonal structure requires whole genome sequencing (WGS). However, given that the samples are from FFPE material, the authors perform whole exome sequencing (WES). This is not ideal especially for liver cancer, which is etiologically associated with hepatitis B and C infection. Viral integration occurs in non-coding as well as coding regions and results in increased rate of copy number changes, which makes WES even less suitable for this kind of analysis.
4. The authors do not perform a statistical analysis of clonality on substitution data. Simply comparing mutation calls in HCC and ICC compartments is not enough to characterize clonal structure. The authors should correct variant allele fraction for copy number and tumor purity and draw a scatter plot of the corrected values for HCC and ICC on different axes. In the absence of such plots it is difficult to interpret the phylogenetic trees. There are public tools to perform such analyses.
5. Separate phylogenetic analyses should be performed on indel and copy number data to assess the validity of the phylogenetic trees drawn using substitutions. This is critical especially when only WES but not WGS is available. The authors identified trunk and branch CNV segments by requiring an overlap of 70%. This is very loose criterion. Proximity of the exact copy number breakpoints or segments needs to be compared maybe within a window of +/- 0.1Mb. Supple. Table 5 gives a list of CNV segments per patient but it is not specified which segments is identified in HCC/ICC.
6. The authors use a method (ref-39) that employs NMF to identify de novo mutational signatures from substitutions and compare the identified signatures to the 30 signatures listed in COSMIC website by cosine similarity. In line 167, they say that they identified six signatures but provide the 96 mutation spectra of three signatures only (Supple. Fig.1). Moreover, they do not indicate which COSMIC signatures these three correspond? Mutation spectra of the reported signatures do not bare resemblance to the COSMIC signatures at least visually.
7. The authors identify viral integration sites from WES data, which is not adequate. Have they done a targeted interrogation of viral integration sites in the samples in which they were not identified? They could perform an analysis of the oncoviral genome by targeted pull-down of the virus sequences associated with liver cancer.
8. The authors perform a statistical analysis to identify genes that are significantly mutation. Is this analysis done on all the mutations identified in a patient and/or separately in HCC and ICC compartments? Is there particular enrichment of certain genes in the trunk vs braches? The authors only list the different pathways significant genes are implicated with. They should give information about what kind of mutations are observed (missense, frameshift, etc) indicating the functional impact and elaborate on how this might relate to clonal structure?
9. Which variable were included in multivariate Cox regression analysis?
10. In line 252, the authors say CNV analysis using WES is highly comparable to that obtained from low-depth WGS data but do not provide any further details.

Reviewer #1

The manuscript (NCOMMS-16-30595-T) is a well-executed manuscript examining mixed hepatocellular-cholangiocarcinoma (H-ChC) genetics. The authors appropriately employ microdissection followed by whole-exome sequencing of the samples. The authors report on differences and similarities on nonsynonymous mutations, somatic copy number variations, and HBV integrations between iCCA, HCC, and H-ChC cancers. The data are novel, contribute to our understanding of these cancers, and the data largely support the conclusions. I have several comments to strengthen the manuscript.

Q: The immunohistochemistry phenotyping of the cancers is incomplete. For HCC, in addition to GPC3, HSP70 and glutamine synthesis should be employed (EASL guidelines Journal of Hepatology 2012). For intrahepatic cholangiocarcinoma, SOX9 should be employed as an additional marker.

R: Thank you very much for your suggestion. Indeed, HSP and SOX9 should be employed in confirm of HCC or iCCA. However, we have no method to get enough sections for the immunohistochemistry due to the lack of enough tissue. We feel really sorry. In our research, the hepatocyte markers (hepatocyte and GPC3) and biliary markers (CK7 and CK19) were used to confirm the diagnosis of HCC and iCCA. All markers are widely used in identifying HCC or iCCA¹⁻⁴. Nearly all samples were diagnosed clinically by experienced pathologists according to the morphology and immunohistochemistry in a golden manner. It is regretful that the additional markers were not employed in our research. In order to acquire the accurate diagnoses, we also re-evaluate all samples by two experienced pathologists of PUCMH according to the 2010 WHO classification. And the proximity of HCC and iCCA components was provided in Figure 2 in our revised manuscript.

Q: The current term for mixed cancers is now Hepato-Cholangiocarcinoma (H-ChC) as defined by the current mixed tumor working group.

R: Thank you very much for your suggestion. We have adopted the term “H-ChC” in our manuscript.

Q: The current term for intrahepatic cholangiocarcinoma is iCCA (see review by Blechacz et al. Nature Reviews in Gastroenterology and Hepatology 2011).

R: Thank you for your advice and we have used iCCA as the term of intrahepatic cholangiocarcinoma in our revised manuscript.

Q: The Kaplan-Meier curve should be deleted from the manuscript. The numbers are too small, and the clinical information to assure equal stratification across the groups is not provided. Hence, all data on patient survival need to be removed from the manuscript.

R: It has to be admitted that the numbers of our study are not big enough. Consequently, the results of survival analyses on these patients may be not very convincing. However, we have no ability to expand our samples due to the rarity of H-ChC. In that case, we further conducted univariate and multivariate analyses on many potential factors in liver cancer. The detailed results are provided in Table 1 and Table 2. And we also discussed them in our manuscript in red.

Q: Telomerase promoter mutations are common in early HCC (Nault JC et al. Nature Genetics 2015). What percent of the mixed tumors had these mutations?

R: After carefully screen for mutations, we found no telomerase promoter mutation in the mixed tumors of our study. It may be attributable to the coverage of WES and the depth of sequencing. Therefore, the high-depth of WGS is more suitable for better telomerase promoter mutations identification.

Q: By an Ingenuity-like Pathway Analysis, were there common pathways driving the differing phenotypes?

R: Through KEGG pathway enrichment on SMGs, we indeed identified some pathways that may drive the differing phenotypes. 12 genes e.g. ACVR2A and APC that constitute signaling pathways modulating pluripotency of stem cells were found to be mutated in 6 H-ChC patients (P1-P6) (Table S13). Moreover, 12 mutated genes in Wnt and Notch pathways including TP53 and NFATC2/3, which regulate the differentiation of hepatocyte and biliary epithelium, were noted in 5 H-ChC patients (P1-P4 and P6) (Table S14). These data suggest the involvement of such key genes and pathway in the differentiation of HCC and iCCA in H-ChC. However, whether there are any corresponding changes in the protein levels of such mutation genes remains unknown, which could be exploited using next-generation RNA sequencing.

Reviewer #2

The authors sequenced 15 combined hepatocellular cholangiocarcinoma, 32 HCC and 28 ICC. WES was performed with DNA extracted from FFPE samples with or without microdissection. Following these analyses, the authors concluded that combined tumors were of clonal origin with substantial intra-tumor heterogeneity.

This is an interesting project, however the manuscript and the experimental design showed major weaknesses and the conclusions are not innovative enough. Here are some selected examples:

Q: Pathological features of these series of samples are not precisely described and this is a major weakness

R: Thank you for your suggestions. Indeed, the pathological features of included H-ChC samples were not described precisely, which may influence the understanding of potential readers. Considering the focus of our research is H-ChC, we provided more detailed information about the proximity of WES H-ChC samples. Therefore, we

provided revised Figure 2 to present the proximity of HCC and iCCA components within H-ChC more clearly. DNA was taken from a single neoplastic mass that contiguously presents HCC and ICC in 4 samples and from physically different foci separated by mesenchymal tissues in 3 samples. And we also discussed the pathological classification of H-ChC. If more detailed information is needed, we will try our best to provide. Thank you very much.

Q: Several clinical data are missing

R: We provided some other important clinical data in our manuscript, which presented in supplementary table 1.

Q: The identified mutations are not sufficiently described in term of putative protein and functional consequences. Allele frequency and coverage should be provided. Consequently the results are very difficult to evaluate with only a list in pdf. Several genes classically mutated in HCC and ICC was not identified in the present work suggesting technical problems in the workflow

R: I feel very sorry for the confusions I have brought to you. And the putative protein and functional consequences of identified mutations in our study have been provided in detail in Supplementary Table 4. The mutations and their corresponding amino acids changes are also provided in this table. In addition, we provided the allele frequency and coverage of identified mutations for each sample in Supplementary Table 4. We also consider the allele frequency and coverage are very important for deep analyses. And I feel very appreciated for your scientific suggestions. Finally, although several genes classically mutated in HCC and ICC indeed were not identified in our work, we also identified several prominent genes such as CDKN2A, APC in HCC components, BAP1 in iCCA components and TP53 in both of these two components. To these results, we think that H-ChC consists of HCC and iCCA components, it is a different type of liver cancer from HCC and iCCA. Although HCC and iCCA components in H-ChC are histological similar to HCC and iCCA, whether they are same with each other is still ambiguous. Furthermore, the WES and the depth

of it may also restrict us from identifying several classically mutated genes in HCC and iCCA.

Q: Nucleotide mutational signatures should be assessed according to COSMIC nomenclature

R: We assessed the nucleotide mutational signatures according to COSMIC nomenclature. The signature A is near to signature 25 identified in Hodgkin lymphomas, however, they did not correspond to each other. The signature B is near to signature 5 found in many types of cancer, however, they are not same. The signature C is near to signature 1 associated with age of cancer diagnosis, however, the filter with cosine similarity >0.9 did not pass. Therefore, the three identified signatures in our research are not similar to the 30 known signatures and they may be associated with some special features of cancer not found.

Q: The text is very difficult to read and to understand.

R: We feel sorry for the convenience. We have tried our best to present our results more clearly and improve the readability of our manuscript.

Reviewer #3

The authors perform a comparative analysis of laser micro-dissected hepatocellular carcinoma (HCC) and cholangiocarcinoma (ICC) components from seven patients diagnosed with combined hepatocellular cholangiocarcinoma, a rare subtype of liver cancer that displays both pathological features. They perform whole exome sequencing to characterize the subclonal structure and determine the relationship between the two components. Although the study contains only seven patients and is underpowered, this is an interesting sample set. They identify common as well as sample-specific somatic alterations. They conclude that the two tumor compartments are of monoclonal origin with the accumulation of substantial sample-specific mutations in the form of branching evolution. Although the inference of the monoclonality is probably true,

there are major flaws in experimental setup as well as the analysis that prevent a true assessment of the level of heterogeneity.

Q: No information about the physical proximity of the HCC and ICC components is given. Were the samples taken from a single neoplastic mass that contiguously presents HCC and ICC or from physically different foci separated by normal-looking liver tissue?

R: We also realized that the pathological features of included H-ChC samples were not described precisely, especially the information about proximity of HCC and ICC components in H-ChC samples. In our revised manuscript, we provided more detailed pathological pictures to present the proximity of all WES H-ChC samples. In our study, we focused on the H-ChC that consists of two distinct liver cancer components. In 7 WES H-ChC samples, 4 samples were taken from a single neoplastic mass that contiguously presents HCC and ICC. In the P2, 3, 4 samples, the samples were taken from different foci separated by mesenchymal tissue rather than normal-looking liver tissue. We provided the detailed histopathology pictures of P2 in Figure 2 to show the proximity of HCC and iCCA components in our revised manuscript. If more detailed histopathological features of all H-ChC samples are needed, please let me know. Thank you very much.

Q: Is there a pathologist's estimate of tumor cellularity? Was cellularity estimated from NGS data? Low tumor purity might affect the identification of mutations in the two histological compartments especially if the coverage is not comparable. Mutation calling algorithms might not be able to identify mutations if the variant allele fraction is low. Have the authors performed a pileup of sample-specific mutations in the other sample to see if there are any aberrant reads supporting the same variant?

R: Firstly, two experienced pathologists help estimate the tumor cellularity of included samples, especially the samples for WES analyses. The seven samples were estimated microscopically and microdissected.

Secondly, thank you for your professional advice again. Indeed, the tumor purity is very important to analyze mutations more accurately. We conducted cellularity estimation from NGS data using ABSOLUTE⁵ that could infer tumor purity and cancer cells ploidy.

Thirdly, through carefully screen on the identified mutational genes, we found that ARID 1B, as the family member of ARID 1A and ARID 2, is frequently mutated in H-ChC samples, which may play an important role in the carcinogenesis of H-ChC. Therefore, we conducted validation in all included H-ChC samples using Sanger sequence. However, no positive result was found due to the low quality and severe degradation of our samples. I admit that our results are not perfect enough and feel regretful for the lack of validation in other samples.

Q: Study of clonal structure requires whole genome sequencing (WGS). However, given that the samples are from FFPE material, the authors perform whole exome sequencing (WES). This is not ideal especially for liver cancer, which is etiologically associated with hepatitis B and C infection. Viral integration occurs in non-coding as well as coding regions and results in increased rate of copy number changes, which makes WES even less suitable for this kind of analysis.

R: We also admit that the WGS is more suitable for clonal study of liver cancer in comparison with WES. Although WGS is preferable, WES is also reliable for mutation analyses and CNV analyses, which is widely accepted in field of liver cancer research⁶⁻⁹.

Q: The authors do not perform a statistical analysis of clonality on substitution data. Simply comparing mutation calls in HCC and ICC compartments is not enough to characterize clonal structure. The authors should correct variant allele fraction for copy number and tumor purity and draw a scatter plot of the corrected values for HCC and ICC on different axes. In the absence of such plots it is difficult to interpret the phylogenetic trees. There are public tools to perform such analyses.

R: Indeed, the copy number and tumor purity could affect the results of clonality analyses. Therefore, we correct variant allele fraction for copy number and tumor purity and draw scatter plot for HCC and ICC on different axes using Pyclone^{10,11}. The results were presented in Figure 5C in our revised manuscript. And we also used the plots to interpret the monoclonality of H-ChC.

Q: Separate phylogenetic analyses should be performed on indel and copy number data to assess the validity of the phylogenetic trees drawn using substitutions. This is critical especially when only WES but not WGS is available. The authors identified trunk and branch CNV segments by requiring an overlap of 70%. This is very loose criterion. Proximity of the exact copy number breakpoints or segments needs to be compared maybe within a window of +/- 0.1Mb. Supple. Table 5 gives a list of CNV segments per patient but it is not specified which segments is identified in HCC/ICC

R: We constructed phylogenetic trees using somatic SNVs and substitutions that take the CNVs in consideration. The results were provided in Figure 5A.

In order to analyze ubiquitous CNV within two tumor components of each H-ChC, we regard an overlap of 70% CNV segments as a ubiquitous CNV according to the criterion of Database of Genomic Variants.

The website: <http://dgv.tcag.ca/dgv/app/faq>

In our revised manuscript, we gave a list of CNV segments for each patient and clearly specified which segments are found in HCC or ICC components in Supplementary Table 5.

Q: The authors use a method (ref-39) that employs NMF to identify de novo mutational signatures from substitutions and compare the identified signatures to the 30 signatures listed in COSMIC website by cosine similarity. In line 167, they say that they identified six signatures but provide the 96 mutation spectra of three signatures only (Supple. Fig.1). Moreover, they do not indicate which COSMIC signatures these three correspond? Mutation spectra of the reported signatures do not bare resemblance to the COSMIC signatures at least visually.

R: In line 167, we say that we identified six signatures but provide the 96 mutation spectra of three signatures only (Supple. Fig.1). We feel very ashamed for the inappropriate and ambiguous expression. The six signatures specified the mutation profiles of bases including C>G/G>C, T>G/A>C, T>A/A>T, T>C/A>G, C>A/G>T and C>T/G>A. And we have corrected the expression in our revised manuscript in red¹². We performed mutation spectrum and signature analysis to explore the relationship within tumor samples in each same patient. We drew mutation spectrum barplot to present mutation spectrum of each sample. We also conducted clustering analysis on mutation spectrum to draw mutation spectrum heatmap to observe the similarity and difference within tumor samples. We conducted cluster analyses on 96 somatic mutation types using Nonnegative Matrix Factorization (NMF)^{13,14} and acquired three different mutation signatures. Then the identified mutation signatures were clustered with 30 known signatures on COSMIC¹⁵ to explain mutation process of samples¹⁶. The similarity of mutation signatures were evaluated with cosine similarity>0.9, which suggested common signatures. Signature A, B and C were identified in tumors samples and the distribution of them were presented (Figure 4A). The signature A is near to signature 25 identified in Hodgkin lymphomas, however, they did not correspond to each other. The signature B is near to signature 5 found in many types of cancer, however, they are not same. The signature C is near to signature 1 associated with age of cancer diagnosis, however, the filter with cosine similarity>0.9 did not pass. Therefore, the three identified signatures in our research are not similar to the 30 known signatures and they may be associated with some special features of cancer not found. Therefore, it is reasonable that the mutation spectra of the reported signatures do not bare resemblance to the COSMIC signatures.

Q: The authors identify viral integration sites from WES data, which is not adequate. Have they done a targeted interrogation of viral integration sites in the samples in which they were not identified? They could perform an analysis of the oncoviral genome by targeted pull-down of the virus sequences associated with liver cancer.

R: In our research, we conducted HBV integration analyses for all WES H-ChC

samples. We mapped the clean data to human and HBV genome using bwa software and extracted the chimeric paired-end reads that some reads mapped to human genome and others to HBV genome. Then these reads were mapped to human and HBV genome using The Basic Local Alignment Search Tool. The sites of human and HBV sequencing integration are breakpoint of HBV integration that supported by at least 2 chimeric paired-end reads. Finally, we conducted annotation of breakpoints using annoVar software. Considering the random features of HBV integration and tens of millions of probable integration sites, the probability of common HBV integration sites in two different origin liver cancer samples could hardly be ignored. Therefore, the common integration supports the monoclonal origin of HCC and iCCA components. However, a lack of common HBV integration site in other samples cannot rule out the possibility of common origin of these two tumor components, especially in light of the data from WES analyses, because HBV could integrate in any sites of tumor genome and the WES data just account for about 1% of whole genome. Furthermore, the HBV integration randomly occurs in any stage of carcinogenesis, either before or after differentiating into distinct phenotypes. Consequently, we indeed questioned the results that viral integration sites in the samples in which they were not identified and we took it reasonable due to the reasons mentioned above. Therefore, we just regard the HBV integration as a supplementary evidence to explore the clonality of H-ChC.

Q: The authors perform a statistical analysis to identify genes that are significantly mutation. Is this analysis done on all the mutations identified in a patient and/or separately in HCC and ICC compartments? Is there particular enrichment of certain genes in the trunk vs braches? The authors only list the different pathways significant genes are implicated with. They should give information about what kind of mutations are observed (missense, frameshift, etc) indicating the functional impact and elaborate on how this might relate to clonal structure?

R: We identified significantly mutation analyses in single patient and separately in HCC and iCCA components. Very few significantly mutations were found in separate

HCC or iCCA compartments. However, we identified 359 significantly mutational genes in all included H-ChC samples. Considering the special features of H-ChC, we speculated that some genes might play an important role in the differentiation of H-ChC into two distinct tumor components. Therefore, we screened SMGs to explore some promising genes and we defined the mutations in at least two samples as significant mutations for special samples. We classified all SMGs into ubiquitous mutations for HCC and iCCA components, HCC and iCCA components. In all 7 H-ChC samples, we identified 14 ubiquitous SMGs including ACACB, ARID1B, CACNG4, CSMD3, FCGBP, FRAM1, GRAM1, MUC12 and so on. Meantime, we found 7 SMGs in HCC samples of H-ChC. They contained ASXL3, C5, FAM205A, KATAP10-2, ZNF468, FMNL2 AND USP2a. For iCCA samples, we also identified 5 SMGs including LRP1B, PCDHGA7, POTEC, SYNE1 and WDR44. For all identified SMGs in special tumor components, we screened the PubMed database and NCBI database to acquire their current research on cancer or differentiation. And we found that many genes play an important role in the carcinogenesis and differentiation of cancer. Similarly, we also conducted identification on driver mutational genes. We found 31 ubiquitous driver mutation genes in all 7 H-ChC patients. 24 and 13 driver mutation genes were identified in HCC and iCCA samples respectively. In our revised manuscript, we give the information about what kind of mutations are observed (missense, frameshift, etc) and added the column of “variant classification” in Supplementary Table12, 13, 14.

Q: Which variable were included in multivariate Cox regression analysis?

R: We feel very sorry for the lack of descriptions for survival analyses and we provided most of important information in our manuscript. Firstly, we conducted univariate analyses on many factors in liver cancer and the results were provided in Table 1 of our revised manuscript. The significant prognostic factors and some important clinical factors such as tumor number, size, vascular invasion and so on were analyzed in multivariate Cox regression analysis. The variables in multivariate Cox regression

analysis were presented in Table 2 of our revised manuscript.

Q: In line 252, the authors say CNV analysis using WES is highly comparable to that obtained from low-depth WGS data but do not provide any further details.

R: Before we conducted CNV analysis using WES, we read many relevant papers. Considering the spectrum of WES, we performed CNV analysis using these data with great cautions. In the article published in *Gastroenterology* in 2016⁹, the author conducted low-depth and exome sequence for CNV in 43 lesions of 10 liver cancer patients and they proved that the CNV analyses using low-depth WGS and WES data are similar. In our manuscript, we cited the research to support the working we had done using WES data. And we did not provide some further details due to the limits of contents.

Reference

1. Roskams, T.A. *et al.* Nomenclature of the finer branches of the biliary tree: canals, ductules, and ductular reactions in human livers. *Hepatology* **39**, 1739-45 (2004).
2. Yeh, M.M. Pathology of combined hepatocellular-cholangiocarcinoma. *J Gastroenterol Hepatol* **25**, 1485-92 (2010).
3. Patsenker, E. *et al.* The alphavbeta6 integrin is a highly specific immunohistochemical marker for cholangiocarcinoma. *J Hepatol* **52**, 362-9 (2010).
4. Zhang, F. *et al.* Combined hepatocellular cholangiocarcinoma originating from hepatic progenitor cells: immunohistochemical and double-fluorescence immunostaining evidence. *Histopathology* **52**, 224-32 (2008).
5. Carter, S.L. *et al.* Absolute quantification of somatic DNA alterations in human cancer. *Nat Biotechnol* **30**, 413-21 (2012).
6. Totoki, Y. *et al.* Trans-ancestry mutational landscape of hepatocellular carcinoma genomes. *Nat Genet* **46**, 1267-73 (2014).
7. Jia, D. *et al.* Exome sequencing of hepatoblastoma reveals novel mutations and cancer genes in the Wnt pathway and ubiquitin ligase complex. *Hepatology* **60**, 1686-96 (2014).
8. Eichenmuller, M. *et al.* The genomic landscape of hepatoblastoma and their progenies with HCC-like features. *J Hepatol* **61**, 1312-20 (2014).
9. Xue, R. *et al.* Variable Intra-Tumor Genomic Heterogeneity of Multiple Lesions in Patients With Hepatocellular Carcinoma. *Gastroenterology* **150**,

- 998-1008 (2016).
10. Roth, A. *et al.* PyClone: statistical inference of clonal population structure in cancer. *Nat Methods* **11**, 396-8 (2014).
 11. Eirew, P. *et al.* Dynamics of genomic clones in breast cancer patient xenografts at single-cell resolution. *Nature* **518**, 422-6 (2015).
 12. Zhang, J. *et al.* Intratumor heterogeneity in localized lung adenocarcinomas delineated by multiregion sequencing. *Science* **346**, 256-9 (2014).
 13. Alexandrov, L.B., Nik-Zainal, S., Wedge, D.C., Campbell, P.J. & Stratton, M.R. Deciphering signatures of mutational processes operative in human cancer. *Cell Rep* **3**, 246-59 (2013).
 14. Gaujoux, R. & Seoighe, C. A flexible R package for nonnegative matrix factorization. *BMC Bioinformatics* **11**, 367 (2010).
 15. Forbes, S.A. *et al.* COSMIC: mining complete cancer genomes in the Catalogue of Somatic Mutations in Cancer. *Nucleic Acids Res* **39**, D945-50 (2011).
 16. Alexandrov, L.B. *et al.* Signatures of mutational processes in human cancer. *Nature* **500**, 415-21 (2013).

Reviewers' comments:

Reviewer #1 (Remarks to the Author):

The manuscript has been rewritten and further data analyzed. By and large, the authors have addressed my previous concerns.

Reviewer #2 (Remarks to the Author):

In this revised version of the manuscript, major concerns remain:

- Lack of precise description of cases, location of microdissection, IHC staining, pictures of the cases, number of nodules. This can impact deeply the findings.
- Mutational data are still impossible to evaluate in the absence of excel file describing precisely all the mutations with in silico analysis of the functional consequences (polyphen or other), the annotation line per line of the patient ID and the sample ID for each patient, the annotation of allele frequency in EXac. In the present file, the table is not directly exploitable by other colleagues. Overall, the number of false positive and false negative mutations is not evaluated.
- Several classical genes are not found mutated and this is a major weakness questioning the methods
- Analysis of mutational signature is not satisfactory: no precise method description, no classical signatures like signature 16 found in all the published HCC papers. The identified signatures are very noisy and should not be interpreted.
- WES, particularly in FFPE samples, is not an appropriate tool to identify HBV integration, these data are not interpretable
- Prognosis results are not novel. The role of EPCAM and CK19 has been well described in HCC in relation with survival.
- Taking into account all these major concerns, I think that conclusions drawn in term of clonal or not clonal mutations are not robust enough.

Reviewer #3 (Remarks to the Author):

The authors have conducted a substantial revision of their analysis and/or the interpretation / tone of their data. A few remaining points are still required to be addressed before this manuscript can be suitable for publication.

1.Regarding the information requested on the physical proximity of the HCC and ICC components the authors have provided some additional information (including figure 2) but this still remains incomplete. in Fig2 the exact foci the samples were taken from? Some patients seem to have more than two foci - The authors should consider a detailed graphical or figure representation of each patient and samples taken for the analysis. Potentially use some of the supplemental material for it.

2.Regarding the estimation of tumour purity and cellularity could the authors provide the pathologist and ABSOLUTE estimates into a supplementary table?

3.The authors should ensure that the limitations of WES as opposed to more comprehensive techniques such as WGS are fully stated in the discussion.

4.In their CNV analysis they authors use dgv to identify ubiquitous segments. Database of genomic variants is a resource to identify genomic regions that undergo structural variation recurrently across different individuals in order to assess the relevance of that locus for disease biology. What the authors are trying to achieve here is to establish clonal relationships between samples from

the same individuals. This means that if a genomic breakpoint is shared, it should be identified at exactly the same position in both samples from the same individual. Given the use of WES for CNV analysis is not ideal, the authors can take a window around the copy number breakpoints/ or segments as requested previously. But checking overlap is not appropriate in this context. To this effect the authors should revise their analysis and data representation.

The authors should provide the protein change of the mutations on the tree representation.

As far as the phylogenetic trees are concerned, 4/7 patients have TP53 mutations on the trunk whilst one patient has ubiquitous CDKN2A and BAP1 mutations. The authors should evaluate whether there is a pattern regarding the type of events and genes that are most frequently represented in the trunk as opposed to the branches and comment on whether any patterns are emerging. (i.e. loss of tumour suppressors as early events in the pathogenesis or late).

The following sentence does not read accurately: Six 168 mutational profiles were observed within tumor samples, in which C: G > T: A 169 dominated. The profile is similar to that of liver cancer summarized by Alexandrov et al 170. The authors should consider revising.

Reviewer #1

The manuscript has been rewritten and further data analyzed. By and large, the authors have addressed my previous concerns.

Reviewer #2 (Remarks to the Author):

In this revised version of the manuscript, major concerns remain:

Q: Lack of precise description of cases, location of microdissection, IHC staining, pictures of the cases, number of nodules. This can impact deeply the findings.

R: We have provided some important information of cases including the tumor numbers, state of vascular invasive, size, satellite lesions and so on in supplementary table 1. The location of microdissection is very complicated. However, we conducted microdissection for HCC or iCCA components by pathological judgment. In another word, we identified the location of microdissection when we ensure the pathological features of tumor components. We provided some pictures of microdissection sections to present the locations (Figure 1). In our revised manuscript, we provided HE and IHC pictures of all 7 H-ChC samples for WES in supplementary figure 5 in red.

Q: Mutational data are still impossible to evaluate in the absence of excel file describing precisely all the mutations with in silico analysis of the functional consequences (polyphen or other), the annotation line per line of the patient ID and the sample ID for each patient, the annotation of allele frequency in EXac. In the present file, the table is not directly exploitable by other colleagues. Overall, the number of false positive and false negative mutations is not evaluated.

R: Indeed, we didn't provide many information about mutational data including the dbSNP ID, AA changes, CpG islands prediction, GO database annotation, KEGG pathway database annotation, PID database annotation and so on. Considering the limitation of content, we will provide our data on special database, which are available to other researchers. In our revised manuscript, we have provided the allele frequency

(FA) in supplementary table 4.

Q: Several classical genes are not found mutated and this is a major weakness questioning the methods.

R: Thank you very much for your professional suggestions. To date, however, we still could not get available data on H-ChC. Therefore, we also could not acquire the classical genes about this special type of liver cancer. Indeed, there are many classical genes such as CTNNB1, ARID1A, PTEN and so on. H-ChC consists of HCC and iCCA components, it is a different type of liver cancer from HCC and iCCA. Although HCC and iCCA components in H-ChC are histological similar to HCC and iCCA, whether they are same with each other is still ambiguous. Furthermore, the WES and the depth of it may also restrict us from identifying several classically mutated genes in HCC and iCCA.

Q: Analysis of mutational signature is not satisfactory: no precise method description, no classical signatures like signature 16 found in all the published HCC papers. The identified signatures are very noisy and should not be interpreted.

R: We performed mutation signature analysis to explore the relationship within tumor samples in each same patient. We conducted cluster analyses on 96 somatic mutation types using Nonnegative Matrix Factorization (NMF) and acquired three different mutation signatures. Then the identified mutation signatures were clustered with 30 known signatures on COSMIC to explain mutation process of samples. The similarity of mutation signatures were evaluated with cosine similarity >0.9 , which suggested common signatures. Signature A, B and C were identified in tumors samples and the distribution of them were presented (Figure 4A). The signature A is near to signature 25 identified in Hodgkin lymphomas, however, they did not correspond to each other. The signature B is near to signature 5 found in many types of cancer, however, they are not same. The signature C is near to signature 1 associated with age of cancer diagnosis, however, the filter with cosine similarity >0.9 did not pass. Therefore, the three identified signatures in our research are not similar to the 30 known signatures

and they may be associated with some special features of cancer not found. Therefore, it is reasonable that the mutation spectra of the reported signatures do not bare resemblance to the COSMIC signatures. Indeed, no classical signature like signature 16 found in HCC was found in our H-ChC samples. According to our understanding on this special type of liver cancer, H-ChC is a different type of liver cancer from HCC and iCCA. Although HCC and iCCA components in H-ChC are histological similar to HCC and iCCA, whether they are same with each other is still ambiguous.

Therefore, it is acceptable for us that no classical mutational signature in HCC was found in our H-ChC. In addition, no unambiguous significance was found on our identified signatures. Therefore, we just gave few interpretations with limitation.

Q: WES, particularly in FFPE samples, is not an appropriate tool to identified HBV integration, these data are not interpretable

R: In our research, we conducted HBV integration analyses for all WES H-ChC samples. We mapped the clean data to human and HBV genome using bwa software and extracted the chimeric paired-end reads that some reads mapped to human genome and others to HBV genome. Then these reads were mapped to human and HBV genome using The Basic Local Alignment Search Tool. The sites of human and HBV sequencing integration are breakpoint of HBV integration that supported by at least 2 chimeric paired-end reads. Finally, we conducted annotation of breakpoints using annovar software. Considering the random features of HBV integration and tens of millions of probable integration sites, the probability of common HBV integration sites in two different origin liver cancer samples could hardly be ignored. Therefore, the common integration supports the monoclonal origin of HCC and iCCA components. However, a lack of common HBV integration site in other samples cannot rule out the possibility of common origin of these two tumor components, especially in light of the data from WES analyses, because HBV could integrate in any sites of tumor genome and the WES data just account for about 1% of whole genome. Furthermore, the HBV integration randomly occurs in any stage of carcinogenesis, either before or after differentiating into distinct phenotypes.

Consequently, we indeed questioned the results that viral integration sites in the samples in which they were not identified and we took it reasonable due to the reasons mentioned above. Indeed, it is not rigorous to identify HBV integration using WES on FFPE samples. However, we just regard the HBV integration as a supplementary evidence to explore the clonality of H-ChC.

Q: Prognosis results are not novel. The role of EPCAM and CK19 has been well described in HCC in relation with survival.

R: I really feel appreciated for your reminder. Indeed, the role of EpCAM and CK19 has been well described in HCC in relation with survival. We conducted survival analysis on EpCAM expression and liver cancer to ensure the completeness of our research. And the results could support our conclusion.

Reviewer #3 (Remarks to the Author):

The authors have conducted a substantial revision of their analysis and/or the interpretation / tone of their data. A few remaining points are still required to be addressed before this manuscript can be suitable for publication.

Q: Regarding the information requested on the physical proximity of the HCC and ICC components the authors have provided some additional information (including figure 2) but this still remains incomplete. in Fig2 the exact foci the samples were taken from? Some patients seem to have more than two foci - The authors should consider a detailed graphical or figure representation of each patient and samples taken for the analysis. Potentially use some of the supplemental material for it.

R: Thank you for your suggestions. We cut 5um thick sections from representative paraffin blocks of H-ChC samples, which consist of unambiguous HCC and iCCA components. Therefore, we just conducted microdissection for one tumor despite of the tumor number in liver. In addition, a single H-ChC tumor often contains more than three morphologically distinct tumor components and classification of these atypical tumors remains problematic. We confirmed the location of microdissection

for HCC or iCCA components by pathological judgment of experienced pathologists. In another word, we identified the location of microdissection when we ensure the pathological features of HCC or iCCA components. We provided some pictures of microdissection sections to present the locations and how our conducted microdissection. In our revised manuscript, we also provided HE and IHC pictures of 7 H-ChC samples for WES in supplementary figure 5 in red.

Q: Regarding the estimation of tumor purity and cellularity could the authors provide the pathologist and ABSOLUTE estimates into a supplementary table?

R: We provided the ABSOLUTE estimation of tumor purity and cellularity in supplementary table 15 in red.

Q: The authors should ensure that the limitations of WES as opposed to more comprehensive techniques such as WGS are fully stated in the discussion.

R: We discussed the limitations of WES on identification of HBV integration and somatic CNVs as opposed to more comprehensive techniques in the part of discussion in red.

Q: In their CNV analysis they authors use dgv to identify ubiquitous segments. Database of genomic variants is a resource to identify genomic regions that undergo structural variation recurrently across different individuals in order to assess the relevance of that locus for disease biology. What the authors are trying to achieve here is to establish clonal relationships between samples from the same individuals. This means that if a genomic breakpoint is shared, it should be identified at exactly the same position in both samples from the same individual. Given the use of WES for CNV analysis is not ideal, the authors can take a window around the copy number breakpoints/ or segments as requested previously. But checking overlap is not appropriate in this context. To this effect the authors should revise their analysis and data representation.

R: Thank your very much for your reminder. Through rigorous consideration and wide

discussion, we also think the CNV analysis in our manuscript is not appropriate. We regarded the exactly same position of both samples from the same individual as common CNV. And we have revised the analysis and data representation in our revised manuscript in red. The ubiquitous somatic CNVs refer to CNVs with 100% overlap in HCC and iCCA components within H-ChC samples. The supplementary table 5 and table 10 have been revised in our revised manuscript in red.

Q: The authors should provide the protein change of the mutations on the tree representation.

R: The amino acid change of the mutations has been provided in supplementary table 4 in detail. And we provided the variant classification of mutations in evolutionary trees in figure 5A like the research published on NEW ENGLAND JOURNAL of MEDICINE by Gerlinger in 2012^[1].

Q: As far as the phylogenetic trees are concerned, 4/7 patients have TP53 mutations on the trunk whilst one patient has ubiquitous CDKN2A and BAP1 mutations. The authors should evaluate whether there is a pattern regarding the type of events and genes that are most frequently represented in the trunk as opposed to the branches and comment on whether any patterns are emerging. (i.e. loss of tumor suppressors as early events in the pathogenesis or late).

R: We identified 51 significantly mutated genes on the trunk of phylogenetic trees. Then we conducted KEGG pathway enrichment analysis using these genes. And we didn't find pathway change with statistical significance (Figure 2). Indeed, TP53 mutations are significantly mutated in H-ChC patients. And it may play an important role in the initiation of pathogenesis. However, verification experiments are needed to confirm it.

Q: The following sentence does not read accurately: Six mutational profiles were observed within tumor samples, in which C: G > T: A dominated. The profile is similar

to that of liver cancer summarized by Alexandrov et al The authors should consider revising.

R: We have revised the sentence in red.

Reference

[1]Gerlinger M, Rowan AJ, Horswell S et al. Intratumor heterogeneity and branched evolution revealed by multiregion sequencing. N Engl J Med 2012; 366: 883-892.

Figure 1: The result of microdissection

Figure 2: The pathway enrichment of significantly mutated genes on the trunk of phylogenetic trees

Statistics of Pathway Enrichment

Reviewers' comments:

Reviewer #2 (Remarks to the Author):

This reviewer is still not satisfied by this revised manuscript. Overall, as previously mentioned in my previous comments, the number of novel results is limited. If we compare with the manuscript published recently by Moeini et al in Journal of Hepatology, here, the number of analyzed sample is lower, the conclusions are unclear regarding the cancer driver genes and comparing to classical HCC and ICCA, innovation in term of subclasses of combined HCC-CC not taken into account. The clonal origin of the mixed tumors already described in Moeini et al etc....

Moreover, the major conclusion of this paper in the abstract is not really founded on the results. Several messages are vague and just derived from raw data.

Reviewer #3 (Remarks to the Author):

The authors seem to have addressed all main points in previous review

Reviewers' comments:

Reviewer #3 (Remarks to the Author):

Whilst the authors have addressed some of the reviewers comments in the response to reviewers a number of issues remain that would be required for this publication. These include:

The authors must provide a table with all the variants identified by sample and fully annotated in regards to the cDNA change, protein change, VAF, transcript annotation, presence in COSMIC, EXAC and effect prediction. This must be provided with this manuscript rather than as a separate dataset.

In regards to mutation signatures the authors do not have sufficient large sample size to perform NMF and the current signature results are questionable. The authors should instead consider to use one of the signature deconvolution packages such as the one provided by Charlie Swanton or the Mutational Pattern package in bioconductor, by performing a supervised analysis of the known 30 mutation signatures.

HPV integration analysis is incomplete and limited with WES and would consider removing or present it as suggestive.

The authors have not included amino acid annotation on the phylogenetic trees in the figure.

Dear editors and reviewers

I really feel grateful that you can give us chance to revise our manuscript. I feel very honorable that you could give comments on our manuscript and we have opportunity to further improve our manuscript. We have tried our best to revise our manuscript point-by-point in red. I will feel very appreciated if we could get the chance to further revise once our manuscript is not eligible. In addition, we are depositing all new data associated with paper in a persistent repository and we will get accession code as soon as possible. Owing to my own reasons, I urgently submit the revised manuscript to you and will send the data accession code to you. Thank you very much for understanding.

Your sincerely

Haitao Zhao

Q: The authors must provide a table with all the variants identified by sample and fully annotated in regards to the cDNA change, protein change, VAF, transcript annotation, presence in COSMIC, EXAC and effect prediction. This must be provided with this manuscript rather than as a separate dataset.

R: In our revised, we provided all the variants identified by sample and fully annotated in regards to protein change, genotype, allelic depths, base quality, depth, allelic frequency, mutation type, presence in COSMIC, ESP6500 in supplementary table 4. However, we did not provide some information including the dbSNP ID, CpG islands prediction, GO database annotation, KEGG pathway database annotation, PID database annotation and so on. Considering the limitation of content, we will provide

our data on special database, which are available to other researchers.

Q: In regards to mutation signatures the authors do not have sufficient large sample size to perform NMF and the current signature results are questionable. The authors should instead consider to use one of the signature deconvolution packages such as the one provided by Charlie Swanton or the Mutational Pattern package in bioconductor, by performing a supervised analysis of the known 30 mutation signatures.

R : A mutation spectrum shows the relative contribution of each mutation type in the base substitution catalogs. The plot spectrum function plots the mean relative contribution of each of the 6 base substitution types over all samples. Error bars indicate standard deviation over all samples. The total number of mutations is indicated (Figure 1-5). The similarity between each mutational profile and each COSMIC signature, can be calculated with `cos_sim_matrix`, and visualized with `plot_cosine_heatmap`. The cosine similarity reflects how well each mutational profile can be explained by each signature individually. The advantage of this heatmap representation is that it shows in a glance the similarity in mutational profiles between samples, while at the same time providing information on which signatures are most prominent. The samples can be hierarchically clustered in `plot_cosine_heatmap` (Figure 6), which is included in our supplementary materials as supplementary figure 3 to support our conclusion. In addition to *de novo* extraction of signatures, the contribution of any set of signatures to the mutational profile of a sample can be quantified. This unique feature is specifically useful for mutational signature analyses

of small cohorts or individual samples, but also to relate own findings to known signatures and published findings. The `fit_to_signatures` function finds the optimal linear combination of mutational signatures that most closely reconstructs the mutation matrix by solving a non-negative least-squares constraints problem. Plot the optimal contribution of the COSMIC signatures in each sample as a stacked barplot (Figure 7). Compare the reconstructed mutational profile of the sample with its original mutational profile (Figure 8). We also can use `ggplot` to make a barplot of the cosine similarities between the original and reconstructed mutational profile of each sample. This clearly shows how well each mutational profile can be reconstructed with the COSMIC mutational signatures. Two identical profiles have a cosine similarity of 1. The lower the cosine similarity between original and reconstructed, the less well the original mutational profile can be reconstructed with the COSMIC signatures. You could use, for example, cosine similarity of 0.95 as a cutoff (Figure 9). However, owing to the difference of named numbers, we presented the samples original numbers and did not correct them. The 002, 005, 006, 007, 008, 009, 010 respectively represent P1, P2, P3, P4, P5, P6 and P7.

Q: HBV integration analysis is incomplete and limited with WES and would consider removing or present it as suggestive.

R: As you suggested, we indeed present it as suggestive. And we have discussed them in the part of discussion in red.

Q: The authors have not included amino acid annotation on the phylogenetic trees in the figure.

R: We have included amino acid annotation on the phylogenetic trees in the figure.

Figure 1

Figure 2

Figure 3

Figure 4

Figure 5

Figure 6

Figure 7

Figure 8 is appended as an individual file

Figure 9

REVIEWERS' COMMENTS:

Reviewer #3 (Remarks to the Author):

The authors have responded with clarity to the reviewer comments. They have specifically extended their supplementary material and included the limitations raised by the reviewers in the discussion. Thank you